# Position: Enough of Scaling LLMs! Lets Focus on *Downscaling*

**Yash Goel** [* 1]   **Ayan Sengupta** [* 1]   **Tanmoy Chakraborty** [1]

## Abstract

We challenge the dominant focus on neural scaling laws and advocate for a paradigm shift toward *downscaling* in the development of large language models (LLMs). While scaling laws have provided critical insights into performance improvements through increasing model and dataset size, we emphasize the significant limitations of this approach, particularly in terms of computational inefficiency, environmental impact, and deployment constraints. To address these challenges, we propose a holistic framework for downscaling LLMs that seeks to maintain performance while drastically reducing resource demands. This paper outlines practical strategies for transitioning away from traditional scaling paradigms, advocating for a more sustainable, efficient, and accessible approach to LLM development.

## 1. Introduction

The development of neural scaling laws has provided a foundational framework for understanding the performance trajectory of large language models (LLMs). These laws (Kaplan et al., 2020; Hoffmann et al., 2022) describe how model performance improves predictably with increases in parameters, dataset size, and compute resources. Initially, scaling laws seemed to offer a clear roadmap for the continuous and predictable advancement of LLMs. However, as the field has evolved — evidenced by the sharp rise in the number of proposed scaling laws between 2020 and 2024, as shown in Figure 1 — significant limitations and challenges have become apparent. These emerging concerns cast doubt on the long-term viability and effectiveness of scaling laws as the primary strategy for advancing AI.

One of the central criticisms of neural scaling laws lies in

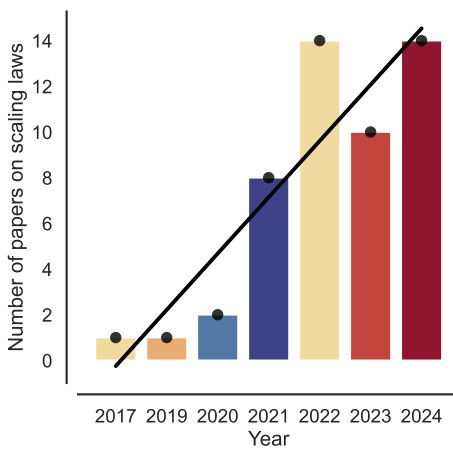

*Figure 1.* The growth of the number of papers on scaling laws for neural models over the past eight years.

their reliance on simplified power law relationships to predict model performance. While these laws capture broad trends, they often fail to account for nuanced factors that influence real-world outcomes. For instance, the assumption that increasing dataset size or compute will indefinitely yield proportional performance improvements ignores diminishing returns observed in practice (Diaz & Madaio, 2024). As models scale, the marginal gains from additional data and compute tend to decrease, leading to inefficient resource allocation (Muennighoff et al., 2023).

Another limitation of neural scaling laws is their emphasis on uniform scaling across model size, data, and compute. Jin et al. (2023) demonstrated that different abilities, such as fact recall and in-context learning, degrade at different rates under pruning or downsizing. This variability suggests that scaling laws may not provide a one-size-fits-all solution and that alternative approaches are needed to address diverse performance requirements. By focusing exclusively on scaling, researchers risk missing opportunities to develop more efficient and specialized models tailored to specific applications.

Neural scaling laws also neglect the broader implications of scaling on energy consumption and environmental sustainability. The computational requirements for training large-scale models are immense, resulting in significant carbon emissions (Faiz et al., 2024). Recent studies showed

---

[*]Equal contribution   [1]Indian Institute of Technology Delhi, India. Correspondence to: Ayan Sengupta <ayan.sengupta@ee.iitd.ac.in>, Yash Goel <ee1210984@ee.iitd.ac.in>.

*Proceedings of the 42nd International Conference on Machine Learning*, Vancouver, Canada. PMLR 267, 2025. Copyright 2025 by the author(s).

that the energy efficiency of compute-intensive tasks varies based on factors such as hardware configuration, server location, and training duration (Zhang et al., 2023). Despite these insights, scaling laws do not incorporate considerations for energy efficiency, leading to strategies that may be technically effective but environmentally unsustainable.

As the limitations of neural scaling laws become more apparent, the development of downscaling laws has emerged as a critical area of focus. These laws aim to understand how to reduce model size, dataset requirements, and compute usage while preserving performance for specific tasks. Downscaling laws are essential for promoting efficiency, sustainability, and accessibility in AI research. **In this position paper, we enlighten on the importance of focused research on downscaling laws and propose strategies for leveraging the insights from neural scaling laws for downscaling LLMs to efficient, environment-friendly and sustainable foundational models.** Smaller models guided by downscaling laws can address resource constraints, enabling broader participation in model development by reducing the financial and computational barriers. Additionally, these laws can provide insights into optimizing task-specific performance, ensuring that models can meet diverse application requirements without excessive scaling.[1]

## 2. Neural Scaling Laws

Neural scaling laws outline how the performance of neural networks improves predictably with increases in model size, data volume, and computational resources. These relationships typically follow power law patterns, providing a framework for optimizing model development.

### 2.1. Scaling Laws of LLMs

The evolution of scaling laws for LLMs began with Kaplan et al. (2020), which established power law relationships (described in Equation 1) between model performance and three key factors: model size, dataset size, and compute. This study revealed that larger models are more sample-efficient, suggesting that optimal compute efficiency can be achieved by training very large models on relatively modest data. Importantly, architectural variations like depth versus width were found to have minimal impact. Subsequent refinements emerged with Hoffmann et al. (2022), known as the *Chinchilla law* (described in Equation 2). This work argued against prioritizing model size alone and demonstrated that balancing model size and training data is critical. For instance, a $70B$ model trained on more data outperformed larger models such as Gopher-$280B$, while using the same compute budget. This insight was further reinforced by Tay

et al. (2023), affirming the superiority of vanilla Transformer architectures over novel designs when scaled, explaining the industry's reliance on standard architectures.

Caballero et al. (2023) introduced the concept of Smoothly Broken Neural Scaling Laws (BNSL), offering a more nuanced framework for understanding scaling behavior. BNSL addressed phenomena like double descent and sharp capability transitions, which traditional scaling laws failed to predict. This work highlighted the limitations of existing models in predicting performance at extreme scales, broadening our understanding of scaling dynamics across domains. In vocabulary scaling, Tao et al. (2024) discovered power law relationship between vocabulary size and model parameters. This work demonstrated that larger models requires proportionally larger vocabularies to optimize performance, challenging existing practices.

---

**Kaplan Scaling Law**

$$L(N, D) = \left[ \left( \frac{N_c}{N} \right)^{\frac{\alpha_N}{\alpha_D}} + \frac{D_c}{D} \right]^{\alpha_D} \tag{1}$$

where $N$ denotes number of model parameters, $D$ denotes the amount of training data, and $N_c$, $D_c$, $\alpha_N$, $\alpha_D$ are fitting constants.

**Chinchilla Scaling Law**

$$L(N, D) = \frac{A}{N^\alpha} + \frac{B}{D^\beta} + E \tag{2}$$

where $A$, $B$, $\alpha$, $\beta$, and $E$ are fitting constants.

The key difference between the two scaling laws is that Kaplan suggested that if the model increases $8\times$, the data set must increase by $5\times$, whereas Chinchilla suggested an equal scaling of $N$ and $D$. Also, Chinchilla proposed an irreducible term in the loss which doesn't vanish on increasing $N$ and $D$.

---

### 2.2. Scaling Laws Beyond LLMs

Studies on scaling small language models have shown promising results. Hu et al. (2024) demonstrated that optimized training strategies could enable small models ($1.2B - 2.4B$ parameters) to rival larger counterparts ($7B - 13B$ parameters). Techniques like the Warmup-Stable-Decay scheduler and extensive use of data ($192\times$ parameter size) improved efficiency and performance, paving the way for deploying models effectively on edge devices.

Hernandez et al. (2021) investigated transfer learning, revealing a power law relationship between pre-training and fine-tuning. Their findings demonstrated that pre-training significantly enhances the utility of smaller fine-tuning

---

[1]The source code of our analysis can be found at https://github.com/LCS2-IIITD/Downscaling.

datasets. However, the phenomenon of "ossification" posed challenges in adapting pre-trained weights, particularly in high-data regimes. Complementary studies (Abnar et al., 2021; Zhang et al., 2024a) highlighted that higher upstream accuracy does not always translate to downstream performance improvements. Notably, the FLP Method (Chen et al., 2024c) merged as a critical advancement, enabling accurate predictions of downstream performance for larger models using smaller-scale data.

Sparse architectures, mainly Mixture of Experts (MoE) models, revolutionized scaling strategies. Clark et al. (2022) highlighted the potential of sparsity in parameter utilization, enabling larger capacities without proportional compute increases. Advances in MoE granularity (Krajewski et al., 2024) refined these models, achieving compute savings up to $40\times$ compared to dense Transformers. Subsequent studies (Yun et al., 2024) balanced expert quantity and inference efficiency, providing a blueprint for optimizing training and deployment.

Inference efficiency has become a focal point of scaling research. Chen et al. (2024a) challenged simple scaling assumptions, showing that sophisticated inference strategies could enable smaller models to outperform larger ones. Techniques like REBASE demonstrated this potential, with Llemma-$7B$ rivaling its $34B$ counterpart at half the computational cost (Wu et al., 2025). Sardana et al. (2024) revealed that smaller models trained longer could meet high inference demands more efficiently. This paradigm shift emphasizes balancing model scale, training duration, and inference strategies to achieve cost-effective deployments.

## 2.3. Criticisms of Neural Scaling Laws

While scaling laws have gained significant popularity, recent studies have highlighted key limitations and uncovered opportunities to refine and extend these frameworks. Diaz & Madaio (2024) critiqued the assumption that larger datasets inherently lead to better model performance. The authors argued that this relationship breaks down when AI systems are deployed to serve diverse human populations. As datasets expand, they increasingly encompass distinct communities with varying, and often conflicting, perspectives on what constitutes a "good" AI system. Current evaluation metrics overlook this diversity, relying on universal benchmarks that may disadvantage certain groups. Villalobos et al. (2024) explored the potential limits of scaling LLMs due to the finite availability of public human-generated text data. It provides a detailed analysis of current trends in the growth of dataset size, the stock of available human-generated data, and the implications for the continued scaling of LLMs. The authors argued that not all human-generated text data is of equal quality, and using low-quality data could hinder model performance. In their analysis, the authors also

included adjustments for data quality and multi-epoch training, revealing that careful filtering and deduplication could enhance the utility of the data stock. However, even with these strategies, they emphasized the need for more efficient data utilization to mitigate the impending data bottleneck. Sorscher et al. (2023) addressed the inefficiency of power law scaling, where marginal performance gains require exponentially more data. The authors proposed data pruning as a solution to achieve exponential improvements in model performance relative to dataset size.

$$CO2eq_{oper} = \sum_i (P_i \cdot eff_i \cdot n_i \cdot t_i) \cdot PUE \cdot c\_inten \tag{3}$$

$$CO2eq_{emb} = \sum_i \frac{t_i \cdot area_i \cdot CPA_i}{lifetime_i} \tag{4}$$

$$CO2eq = CO2eq_{oper} + CO2eq_{emb} \tag{5}$$

where $CO2eq_{oper}$ and $CO2eq_{emb}$ denote operational and embodied carbon footprints, respectively; $P_i$, $eff_i$, $n_i$, $t_i$, $area_i$, $CPA_i$, and $lifetime_i$ denote the peak power, hardware efficiency, count, execution time, area, carbon emitted per unit area and lifetime of hardware unit $i$, respectively; $PUE$ denotes power usage efficiency of the specific data center; and $c\_inten$ denotes the carbon intensity of the specific data center.

**Proposition 2.1.**

$$CO2eq(P, D) = (K_1 + K_2) \cdot N \cdot D \tag{6}$$

*where $(K_1 + K_2)$ represents a compound constant that encapsulates all hardware, data center, and efficiency parameters. This shows that CO2 emissions scale linearly with both (i) the number of model parameters $(N)$, and (ii) the amount of training data $(D)$.*

**Computational Costs of Scaling Laws.** LLMs require significant computational resources for training, inference, and storage, leading to substantial carbon emissions. These emissions can be divided into operational costs, stemming from energy usage during training and deployment, and embodied costs, which result from the manufacturing of hardware such as GPUs, TPUs, and storage devices. To address the growing computational costs of neural scaling, frameworks like LLMCarbon (Faiz et al., 2024) (See Appendix A for details) provide valuable tools for estimating and optimizing carbon footprints. We describe the estimated operational, embodied and total carbon footprint of an LLM pre-training in Equations 3, 4 and 5, respectively.

Further simplification, as shown in Equation 6, reveals that carbon footprints scale linearly with both LLM size and

pre-training token size[2]. We empirically validate the scaling laws of carbon cost for different model and pre-training data sizes, as illustrated in Figures 2 and 3, respectively. While Kaplan scaling law suggests a logarithmic scaling of test loss with respect to model size ($L \sim N^{-0.08}$) and pre-training token size ($L \sim D^{-0.1}$), carbon cost increases linearly with both the factors. Moreover, both Kaplan and Chinchilla scaling laws suggest that test loss $L > \max(N^{-\alpha}, D^{-\beta})$, for some arbitrary constants, $\alpha$ and $\beta$. Therefore, the test loss of an LLM is always irreducible, *i.e.,* can not be brought down below a theoretical bound, irrespective of the model or pre-training data size.

By combining the Kaplan scaling law (Equation 1) with our $CO_2$ emission formula (Equation 6), we can establish a direct relationship between model performance and environmental impact: From Kaplan $L(N) \propto N^{-\alpha}$; where L is the loss and N is the number of parameters, with $\alpha \approx 0.08$ . Solving for N in the Kaplan equation and substituting, we can express $N \propto L^{-1/\alpha}$. Therefore putting in Equation 6:

$$CO_2eq \propto L^{-1/\alpha} \qquad (7)$$

Recent work by Chen et al. (2025) has shown that downstream performance P can be modeled as:

$$P = w_1 + w_2 \cdot L \qquad (8)$$

Where $w_1$ and $w_2$ are task-specific constants. This gives us:

$$CO_2eq \propto (P)^{-1/\alpha} \qquad (9)$$

Given that $\alpha \approx 0.08$, we can approximate:

$$P \propto CO_2eq^{\alpha} \approx CO_2eq^{0.08} \qquad (10)$$

This demonstrates that performance improvements scale approximately with the 0.08 power of carbon emissions. Put differently, to achieve linear improvements in performance, carbon emissions must increase exponentially - a 10% improvement in performance would require approximately $1.1^{12.5} = 329\%$ more carbon emissions. Moreover, as the model's performance increases, improving it further requires exponentially more computing.

## 3. Rise of Small and Efficient Language Models

Several converging practical and technical factors have driven the rise in popularity of Small Language Models (SLMs), which are popular for their size, efficiency, and ability to maintain competitive performance while offering cost-effective and resource-efficient solutions. Studies indicate that models like TinyLlama (Zhang et al., 2024b) (1.1B

[2]The proof is provided in Appendix B.

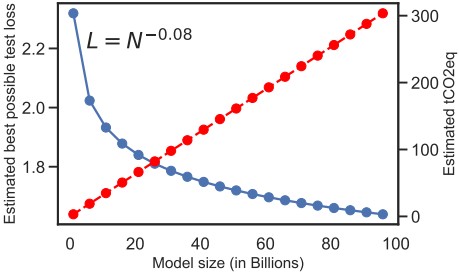

*Figure 2.* Test loss decreases logarithmically with model size, whereas the estimated training-time carbon emission increases linearly.

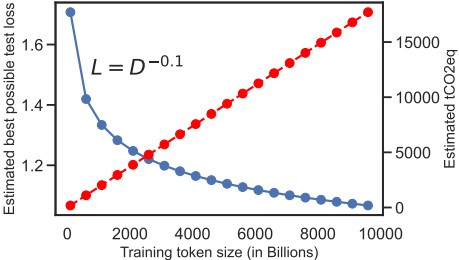

*Figure 3.* Test loss decreases logarithmically with training token size, whereas the estimated training-time carbon emission increases linearly.

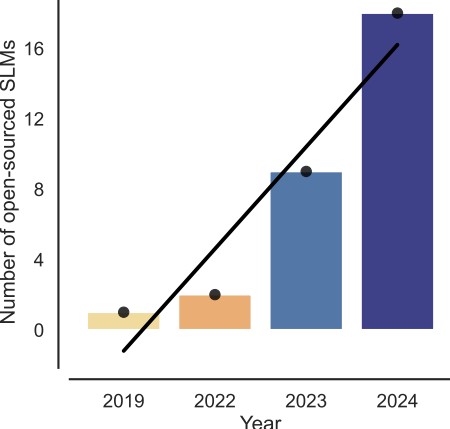

*Figure 4.* Number of open-sourced SLMs (size $100M$-$5B$) developed over the years (statistics taken from Lu et al. (2024b)).

parameters), Mistral-$7B$ (Jiang et al., 2023a), and Phi-4 (Abdin et al., 2024) ($14B$ parameters) can rival larger models in performance despite their more compact architectures (Lu et al., 2024a).

Figure 4 highlights the sharp rise of SLM developments between 2023 and 2024. The momentum behind SLMs has been further accelerated by their practical advantages in real-world applications. The ability to run these models on consumer hardware, from laptops to edge devices, has made them particularly attractive for businesses and developers who need to balance performance with resource constraints. For instance, models like Gemini Nano (Team

et al., 2024) have been specifically designed for mobile devices. In contrast, others like Code Llama (7B) (Rozière et al., 2024) have demonstrated that smaller models can excel in specialized tasks like code generation. Additionally, the cost-effectiveness of training and deploying SLMs, with their faster inference times and lower energy consumption, has made them increasingly appealing in an era where environmental impact and operational efficiency are key considerations. The success of open-source SLMs has also fostered a vibrant community of researchers and developers, who continue to push the boundaries of what is possible with smaller architectures. SLMs have emerged as strong competitors to larger models due to several defining characteristics:

- **High-quality, domain-specific training data.** Rather than relying solely on scale, SLMs prioritize data, emphasizing reasoning patterns and domain relevance.
- **Efficient attention mechanisms.** Techniques like grouped-query attention (GQA) and sliding window attention (SWA) capture long-range dependencies with minimal computational overhead.
- **Targeted architectural optimizations.** Custom architectures tailored for specific domains boost performance.
- **Knowledge distillation.** Advanced methods effectively transfer knowledge from larger models to smaller ones.
- **Model compression techniques.** Sophisticated quantization and pruning strategies reduce model size while preserving capabilities.

### 3.1. Data Quality Evolution

The early success of SLMs revealed that data quality can be more crucial than quantity. Recent models have demonstrated diverse approaches to data quality optimization. Phi-4 (Abdin et al., 2024) introduces a sophisticated multi-layered approach featuring plurality-based filtering, where multiple generated answers per question help filter out overly simplistic or ambiguous content, along with a self-revision mechanism that enables continuous improvement of training data quality. QWEN 2.5 (Qwen et al., 2025) balances scale with quality by implementing rigorous filtering through its instruction models while strategically integrating domain-specific datasets for mathematics and coding to achieve state-of-the-art performance in these areas. MobileLLaMA (Chu et al., 2023) demonstrates the effectiveness of combining high-quality multimodal datasets with supervised fine-tuning using carefully curated dialogue data. These approaches collectively highlight a shift from pure scale to sophisticated data curation techniques, suggesting that the future of language models depends more on intelligent data processing than model size alone.

### 3.2. Architectural Innovations

Several key architectural innovations have enabled SLMs to achieve impressive efficiency. Models like Mistral-$7B$ (Jiang et al., 2023a) introduce improvements such as grouped-query attention (GQA) and sliding window attention (SWA), allowing them to outperform larger parameter models in specific tasks. Models like Mamba (Gu & Dao, 2025) use hybrid architectures, combining transformers with state space models, providing new approaches to achieving efficiency while maintaining performance. Also, the development of efficient attention mechanisms and optimization techniques has allowed models like TinyLlama (Zhang et al., 2024b) to achieve high training throughput while maintaining small parameter counts.

### 3.3. Advancements in Task Specialization and Training Methodology

The field has seen significant progress in developing specialized SLMs that excel in specific domains. For example, WizardMath (Luo et al., 2025) demonstrates strong mathematical reasoning capabilities and surpasses most open-source models with the size ranging between $7B$ to $40B$. Similarly, WizardCoder (Luo et al., 2024) shows that smaller models could outperform larger ones in code generation. Advanced knowledge distillation techniques and targeted training approaches have enabled this specialization. The evolution of training methodologies has been crucial in the evolution of SLMs. As demonstrated by the Orca (Mukherjee et al., 2023) models, progressive learning approaches show how smaller models could effectively learn from larger ones through carefully structured training processes. Despite their smaller size, explanation tuning and chain-of-thought approaches have enabled SLMs to develop stronger reasoning capabilities. The development of more sophisticated fine-tuning techniques has allowed for better transfer of capabilities from larger to smaller models.

### 3.4. Efficiency Optimizations

Post-training optimizations have become increasingly sophisticated. Advanced quantization techniques like SmoothQuant (Xiao et al., 2023) have enabled significant model compression without substantial performance loss. Structured pruning approaches have allowed for systematic model size reduction while preserving key capabilities. Developing efficient inference techniques has made SLMs more practical for real-world deployment. These developments suggest that SLMs will continue to close the gap with larger models while offering significant advantages in terms of efficiency and practicality.

# 4. Our Recommendations

The conventional wisdom of neural scaling laws — that bigger models and more data inevitably lead to better performance, is fundamentally flawed and computationally inefficient. While recent LLMs have demonstrated impressive capabilities through scaling, this success comes at exponentially increasing computational and environmental costs. In this section, we describe our position on **downscaling large language model with domain adaptation** (our specific recommendations are highlighted with underlined text).

## 4.1. Downscaling Datasets for Better Knowledge Alignment

The wide success of SLMs has demonstrated the power of cleaned and high quality data. Also, the evidence from data pruning methods (Sorscher et al., 2023) demonstrates that strategic selection of training examples can achieve exponential rather than power law scaling, fundamentally undermining the assumption that we need ever-increasing amounts of data. This is especially important in light of findings from data-constrained scaling (Muennighoff et al., 2023), which demonstrate that in scenarios with limited data, training smaller models for more epochs outperforms increasing model size, challenging previous scaling assumptions. Future research should focus on optimizing the utilization of existing data over merely accumulating larger datasets.

The emergence of the Domain-Continual Pre-Training (D-CPT) law (Que et al., 2024) (see Appendix C for details) provides a systematic framework to find optimal mixture ratios between general and domain-specific data for domain adaptation, to counter catastrophic forgetting of general data. It eliminates wasteful trial-and-error approaches, achieving remarkable accuracy across six diverse domains. This scientific approach to model development offers a more sophisticated alternative to brute-force scaling.

## 4.2. Model Downscaling

A branch of research has focused on compressing LLMs to reduce computational and hardware requirements using various pruning techniques. Unstructured pruning (Sun et al., 2023) removes individual weights, producing sparse matrices that maintain performance but are less hardware-efficient. Semi-structured pruning (Frantar & Alistarh, 2023), such as the 2:4 sparsity pattern (Pool et al., 2021), introduces a hardware-friendly structured sparsity that accelerates computation. Structured pruning (Ashkboos et al., 2024; Yuan et al., 2023; Sengupta et al., 2025) takes a broader approach by removing entire components, such as Transformer layers (depth pruning) (Fan et al., 2019) or reducing embedding dimensions and attention heads (width

pruning) (Zhu et al., 2021). After pruning, post-training is crucial to mitigate performance degradation. This involves fine-tuning or continual pre-training on datasets tailored to enhance performance recovery while maintaining efficiency. The P2 law (Chen et al., 2024b), highlighted in Equation 11, provides a predictive framework for the post-training loss, considering factors such as pruning rate, model size, pre-pruning loss, and the number of training tokens. This law enables to balance computational costs with performance recovery by identifying optimal post-training data sizes. While higher pruning rates inevitably lead to larger initial losses, effective post-training strategies significantly minimize this impact, allowing smaller models to perform on par with their larger counterparts in many scenarios.

$$
\begin{aligned}
L(N_0, D, \rho, L_0) = \\
L_0 + \left(\frac{1}{\rho}\right)^\gamma \left(\frac{1}{N_0}\right)^\delta \left(\frac{N_C}{N_0^\alpha} + \frac{D_C}{D^\beta} + E\right) \quad (11)
\end{aligned}
$$

where $L_0$ is the uncompressed model loss, $\rho$ is the pruning rate, $N_0$ is the model size before pruning, $D$ is the number of post-training tokens, and $N_C, D_C, E, \alpha, \beta, \gamma$ are fitting constants.

## 4.3. Ensemble of LLMs

The recent advancements in model merging (Yang et al., 2024) provide an exciting opportunity to combine multiple trained models in an efficient and effective manner. This offers significant practical benefits, including reduced storage and inference costs, while also preserving privacy by eliminating the need for original training data. Although model merging is one way to integrate LLM capabilities, ensemble approaches have become more popular because they offer more consistent performance increases and greater flexibility. Ensemble approaches can effectively exploit diverse LLMs despite of their architectural differences, while merging needs models to share compatible architectures and parameter spaces and does not guarantee better performance after combination. Ensemble techniques are especially appealing for practical applications because of their flexibility and the fact that they tend to produce more consistent performance improvements.

Lu et al. (2024a) categorized LLM ensemble strategies into three main approaches: (i) before inference through router-based model selection, (ii) during inference via token-level integration, and (iii) after inference through output combination or selection. Pre-inference ensemble methods employ routers to select the most appropriate LLM for a given input, optimizing for both performance and computational efficiency. For example, ZOOTER (Lu et al., 2023) uses a reward model to train a router that selects the optimal LLM based on query characteristics. During-inference ensem-

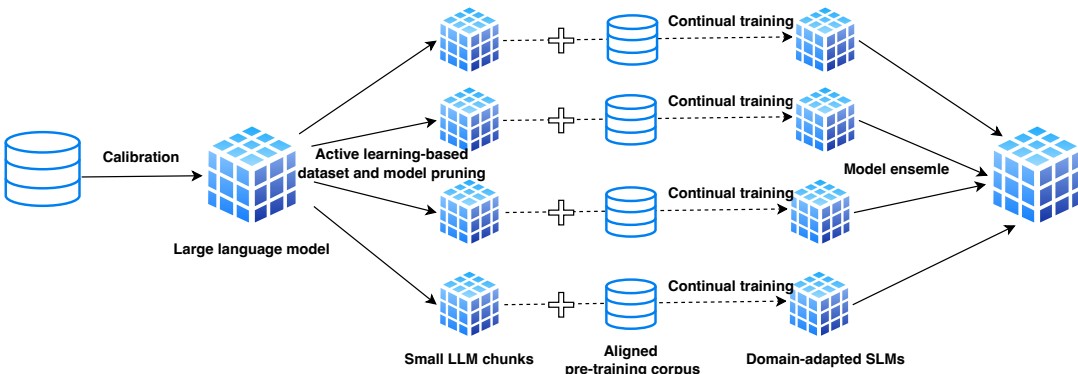

*Figure 5.* Our proposed framework for model downscaling with adaptive domain adaptation. We decompose an LLM pre-training into multiple stages – (1) calibration for compressing LLM into multiple SLM chunks, along with aligned pre-training corpus, (2) active learning-based dataset pruning, for optimal data downscaling, (3) Continual training of SLMs on aligned corpus and (4) model ensemble for merging SLM chunks to recover the larger model with original parameter size.

ble approaches operate at a finer granularity by combining outputs at each decoding step (Li et al., 2024), which can help reduce exposure bias and hallucinations. However, these methods often face challenges with heterogeneous vocabularies across different LLMs (Xu et al., 2024). Post-inference ensemble strategies combine multiple generated outputs or build model cascades where smaller models handle simpler queries and larger models are invoked only when necessary. This hierarchical approach, as demonstrated by Chen et al. (2023), can maintain quality while significantly reducing computational costs, showcasing how ensemble methods can deliver practical benefits that are harder to achieve through merging alone.

$$L(L_0, n) = L_0 - b + \frac{b}{n^a} \tag{12}$$

where $L_0$ is the base model loss, $n$ is the number of models in the ensemble and $a, b$ are fitting constants.

Lobacheva et al. (2021) introduced scaling law of deep ensemble, which can be analysed as shown in Equation 12 (see Appendix D for more details), which describes the expected ensemble loss as the number of models in the ensemble increases. Crucially, the authors revealed a "Memory Split Advantage" effect, where using multiple smaller networks can outperform a single large network for a given computational budget. Although Lobacheva et al. (2021) investigated the law primarily for CNN architectures, the key insights found in the study can be extended to other neural architectures as well. This finding suggests that practitioners can improve model performance by strategically splitting their computational resources across several medium-sized network models rather than investing in one large model.

## 4.4. Why Downscaling Law Might Work?

As we have all the necessary tools, such as post-training after pruning, and expected loss of ensemble, we can combine the knowledge from Equation 11 and Equation 12, to propose Equation 13 (see Appendix E for details), establishing a robust theoretical guarantee of efficient downscaling. It describes a condition where increasing the number of models in a deep ensemble can lead to lower expected ensemble loss $L(L_0, n)$ compared to the base model loss while maintaining the same overall computational cost. The key insight is that by increasing the number of models, as long as the condition in Equation 13 is satisfied, the ensemble can capture more diversity and reduce the overall loss. This is a noteworthy finding as deep ensembling enhances model performance without the need for additional computational resources.

**Proposition 4.1.** *For $n \in \mathbb{Z}_+$ satisfying*

$$\left( \frac{n^a - 1}{n^{a+\gamma}} \right)(n-1)^\gamma \geq \frac{1}{bN_0^\delta} \tag{13}$$

*The expected ensemble loss $L < L_0$, the loss by the base model, at the same computational cost $C$.*

To show the implication of the equation in depth, let us take an example of the LLaMA-3-8$B$ model. From Chen et al. (2024b), we obtain $\alpha = -1.57$ and $\gamma = 1.08$. Similarly, taking some loose assumptions from Lobacheva et al. (2021), we obtain $a = 0.83, b = 0.83, \delta = 0.29$. Putting these values in Equation 13, it holds $\forall n > 7$. This implies that we can use 8 or more pruned versions of LLaMA-3-8$B$, each with $1B$ parameters, in an ensemble that would outperform the original model without incurring any additional cost.

## 4.5. Proposed Downscaling Pipeline

Integrating these insights into model efficiency, scaling laws and benefits of SLMs, we propose a novel pipeline for developing more efficient and domain-adapted language models, as shown in Figure 5. This pipeline leverages techniques from dataset optimization, model compression, and ensemble methodologies to create smaller, more efficient, yet competent language models.

The first phase employs active learning-based techniques for dataset pruning (Sorscher et al., 2023) and model pruning. Using Bayesian active learning strategies (Bayer & Reuter, 2024), we identify the most informative samples from the training data while simultaneously determining optimal model architectures using unstructured, semi-structured pruning, or structured pruning (Sun et al., 2023; Frantar & Alistarh, 2023; Ashkboos et al., 2024; Yuan et al., 2023; Sengupta et al., 2025). This dual optimization approach allows us to create multiple smaller, more efficient model variants from a single LLM while maintaining essential capabilities. Also, we obtain a collection of high quality and carefully-curated datasets. The data pruning process is guided by theoretical frameworks such as Sorscher et al. (2023), suggesting that strategic selection of samples can help break the power law. The pruning of models should be done in a manner that it follows our proposition of downscaling (Equation 13).

These pruned models are designed to specialize in specific domains using domain continual pre-training. The parallel streams shown in the pipeline (Figure 5) represent these different SLM variants, each paired with an aligned pre-training corpus carefully curated for their target domain. This is crucial for efficient specialization and follows the D-CPT law for systematic domain adaptation, without catastrophic forgetting, and guided by theoretical frameworks such as the P2 law, which helps predict and minimize post-compression performance degradation. This allows each model to develop deep expertise in its designated area while maintaining computational efficiency. This targeted training approach eliminates wasteful trial-and-error methods typically associated with domain adaptation, as each model receives only the most relevant data for its specialization.

The final stage of our pipeline implements a model ensemble methodology to combine these domain-adapted SLMs into a unified system. This ensemble approach preserves the specialized capabilities of individual SLMs while creating a more versatile final model. By leveraging advanced ensemble techniques (He et al., 2020; Yadav et al., 2024), we can effectively aggregate the domain expertise of each component model while maintaining a smaller computational footprint compared to traditional large-scale models. This approach is fundamentally different from mixture of experts, where the primary motivation is usually to encour-

age sparsity within different experts. Rather, we focus on combining multiple smaller experts together in a more collaborative manner, ensuring better combined performance. As shown earlier in Proposition 4.1, this ensemble would give a better performance than the original model at the same computational cost.

The success of this pipeline relies on careful orchestration between components and systematic empirical validation. Our approach addresses three critical challenges in modern language model development: reducing computational requirements through strategic pruning, maintaining performance through targeted domain adaptation, and combining specialized capabilities through sophisticated ensemble methods. The result is a more efficient, domain-aware language model that achieves high performance without excessive computational demands or model scale.

## 5. Alternative Views

### 5.1. Limitations of Downscaling

Despite their wide success, SLMs frequently have difficulties with complex language understanding and contextual intricacies. Also, they may not attain the same degree of precision in multifaceted reasoning or complex data patterns as their larger counterparts which leads to diminished performance on activities necessitating profound comprehension or considerable expertise across multiple fields. Furthermore, Jin et al. (2023) revealed crucial insights into how language model capabilities degrade during downscaling, challenging the assumption that reduction in model size uniformly affects all capabilities. Their findings demonstrate a stark contrast in how different cognitive abilities deteriorate. While fact recall begins to significantly degrade when more than 30% of weights are removed through pruning, in-context learning capabilities show remarkable resilience, maintaining performance even with aggressive pruning of up to 60-70% of weights. This pattern remains consistent across various model reduction approaches, including pruning and dense scaling, and has been observed across different model families like OPT and LLaMA. This suggests that the neural mechanisms responsible for storing factual knowledge require significantly more parameters than those supporting in-context learning capabilities. This insight improves SLM deployments by tailoring model reduction strategies to the target application's fact recall versus in-context learning needs.

### 5.2. Limitations of Ensembling

Despite their advantages, ensemble techniques for LLMs face several significant limitations that affect their practical deployment and effectiveness. As highlighted in the survey by Lu et al. (2024a), each ensemble approach has

its own set of challenges. Pre-inference ensemble methods heavily rely on the accuracy of routing mechanisms, and the router's ability to generalize to new queries (Shnitzer et al., 2023). During-inference ensemble methods struggle with heterogeneous model architectures and vocabularies, making it challenging to directly combine outputs from different LLM families (Huang et al., 2024). Post-inference ensemble approaches face challenges related to the quality and diversity of candidate pools (Jiang et al., 2023b). Additionally, they are particularly resource-intensive as they require both multiple model executions and additional computation for output selection or fusion. A practical limitation across all ensemble approaches is the increased latency in response generation, which can be particularly problematic in real-time applications where quick responses are crucial.

## 6. Conclusion

In this position paper, we critiqued the underlying flaws of neural scaling laws and presented proof that the Carbon Dioxide emissions of training LLMs scale linearly with both the number of parameters and dataset size. This indicates that the scaling of LLMs is not sustainable in the long term, given the growing emphasis on climate change. Our study encouraged exploration of downscaling laws, where instead of training larger models on larger datasets, we proposed training smaller models on aligned pre-training datasets for robust performance without introducing additional computational costs. Utilising insights from recent research, we offered a downscaling law that estimates the number of smaller models in an ensemble that may outperform the original model. The study paves way for more sustainable approaches for training language models.

## Impact Statement

This paper presents work whose goal is to advance the field of Machine Learning. There are many potential societal consequences of our work, none which we feel must be specifically highlighted here.

## Acknowledgment

T. Chakraborty acknowledges the support of the IBM-IITD AI Horizons network and Rajiv Khemani Young Faculty Chair Professorship in Artificial Intelligence.

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

## A. CO2 Emission Formulas from LLMCarbon ([Faiz et al., 2024](#))

**FLOP Count Formulas**:

$$\text{Training: } TC \approx 6ND \tag{14}$$

$$\text{Inference: } IC \approx 2ND \tag{15}$$

where $N$ denotes number of parameters, and $D$ denotes amount of training data.

**Device Execution Time**:

$$t_{dev} = \frac{TFLOP}{n_{dev} \cdot FLOP_{peak} \cdot eff} \tag{16}$$

where $TFLOP$ denotes FLOP count, $n_{dev}$ denotes number of computing devices, $eff$ denotes the hardware efficiency, and the computing device number, and $FLOP_{peak}$ represents the device peak throughput.

**Hardware Energy**:

$$energy_{hard} = \sum_{i \in hardware\_set} (P_i \cdot eff_i \cdot n_i \cdot t_i) \tag{17}$$

where $P_i, eff_i, n_i, t_i$ denote the peak power, hardware efficiency, count, and execution time of hardware unit $i$, respectively.

**Operational Energy**:

$$energy_{oper} = energy_{hard} \cdot PUE \tag{18}$$

where $PUE$ denotes Power Usage Efficiency of the specific data center.

**Operational carbon footprint**:

$$CO2eq_{oper} = energy_{oper} \cdot c\_inten \tag{19}$$

where $c\_inten$ denotes the carbon intensity of the specific data center.

**Embodied Carbon Footprint**:

$$CO2eq_{chip} = area \cdot CPA \tag{20}$$

$$CO2eq_{emb} = \sum_{i \in hardware\_set} \frac{t_i \cdot CO2eq_{chip_i}}{lifetime_i} \tag{21}$$

where $CO2eq_{chip}$ denotes chip's embodied carbon footprint, $CPA$ denotes Carbon emitted Per unit Area and $lifetime$ denotes lifetime of hardware unit.

**Total Carbon Footprint**:

$$CO2eq = CO2eq_{oper} + CO2eq_{emb} \tag{22}$$

## B. Proof of Proposition 2.1

Equation 16 can be expressed in terms of Equation 14:

$$t = \frac{6ND}{n \cdot FLOP_{peak} \cdot eff} \text{ [for training]} \tag{23}$$

Starting with the basic formula and substituting each component:

$$CO2eq_{oper} = energy_{oper} \cdot c\_inten \qquad \text{(from 19)}$$

$$= (energy_{hard} \cdot PUE) \cdot c\_inten \qquad \text{(from 18)}$$

$$= \left( \sum_i P_i \cdot eff_i \cdot n_i \cdot t_i \right) \cdot PUE \cdot c\_inten \qquad \text{(from 17)}$$

$$= \left( \sum_i P_i \cdot eff_i \cdot n_i \cdot \frac{6ND}{n \cdot FLOP_{peak} \cdot eff_i} \right) \cdot PUE \cdot c\_inten \qquad \text{(from 23)}$$

Similarly for embodied CO2 from Equation 21 and Equation 23:

$$CO2eq_{emb} = \sum_i \frac{6PD}{n \cdot FLOP_{peak} \cdot eff_i} \cdot \frac{area_i \cdot CPA_i}{lifetime_i}$$

Combining operational and embodied CO2:

$$CO2eq(N, D) = CO2eq_{oper} + CO2eq_{emb}$$

Grouping terms with P and D and further simplifying:

$$CO2eq(N, D) = (K_1 + K_2) \cdot N \cdot D \tag{24}$$

where

$$K_1 = \left( \sum_i P_i \cdot eff_i \cdot n_i \cdot \frac{6}{n \cdot FLOP_{peak} \cdot eff_i} \right) \cdot PUE \cdot c\_inten$$

$$K_2 = \sum_i \frac{6}{n \cdot FLOP_{peak} \cdot eff_i} \cdot \frac{area_i \cdot CPA_i}{lifetime_i}$$

$K_1 + K_2$ represents a compound constant that encapsulates all hardware, data center, and efficiency parameters.

## C. Domain-Continual Pre-Training Scaling Law

Que et al. (2024) provided a mathematical framework to determine the optimal mixture ratio between general and domain-specific data during continual pre-training. It eliminates the need for expensive grid search experiments to find the best ratio and enables more efficient resource allocation in domain adaptation of LLMs. They validated its effectiveness across code, math, law, chemistry, music and medical domains. The law is as follows:

$$L(N, D, r) = E + \frac{A}{N^\alpha} + \frac{B \cdot r^\eta}{D^\beta} + \frac{C}{(r + \epsilon)^\gamma} \tag{25}$$

where $N$ denotes number of parameters, $D$ denotes dataset size, $r$ denotes mixture ratio and $E, A, B, C, \alpha, \beta, \gamma, \eta, \epsilon$ are fitting parameters.

## D. Proof of Equation 12

The scaling law of model ensemble proposed by Lobacheva et al. (2021) is:

$$L = c + \frac{b}{n^a} \tag{26}$$

where $L$ is the ensemble loss, $n$ is the number of models in the ensemble, and $c, b$ are fitting parameters.

For getting a single base model loss $L_0$, we can put $n = 1$:

$$L_0 = L(1) = c + b \implies c = L_0 - b$$

Putting this back in Equation 26, we get our final form:

$$L = L_0 - b + \frac{b}{n^a}$$

## E. Proof of Proposition 4.1

For P2 scaling law in Equation 11, we get

$$L_{\text{compressed}} = L_0 + \left( \frac{1}{\rho} \right)^\gamma \left( \frac{1}{N_0} \right)^\delta \left( \frac{N_C}{N_0^\alpha} + \frac{D_C}{D^\beta} + E \right) \tag{27}$$

$$\approx L_0 + \frac{1}{\rho^\gamma} \cdot \frac{1}{N_0^\delta} \tag{28}$$

For simplicity, we assume the term $\left( \frac{N_C}{N_0^\alpha} + \frac{D_C}{D^\beta} + E \right)$ remains approximately constant when comparing the base model and the ensemble model at the same computational budget. This is justified as both the original model and ensemble use similar total parameter counts and similar computational resources, merely distributed differently.

Therefore, this term appears in both sides of our inequality and cancels out, allowing us to focus on the essential components of the inequality that determine when the ensemble outperforms the base model.

We can know that $\rho \to$ pruning rate $= \left( \frac{n-1}{n} \right)$.

The ensemble loss is given by Equation 12:

$$L_{ensemble} = L - b + \frac{b}{n^a} \tag{29}$$

We need $L_{ensemble} \le L_0$. So,

$$L_0 + \frac{1}{\rho^\gamma N_0^\delta} - b + \frac{b}{n^a} \le L_0 \tag{30}$$

This implies:

$$\frac{1}{\delta^\gamma N_0^\delta} - b + \frac{b}{n^a} \le 0 \tag{31}$$

Substituting $\rho = \frac{n-1}{n}$ and rearranging:

$$\left( \frac{n^a - 1}{n^{a+\gamma}} \right) (n-1)^\gamma \ge \frac{1}{bN_0^\delta} \tag{32}$$

The computational cost of the ensemble is $n \cdot K \cdot (1 - \rho) \cdot N \cdot D = K \cdot N \cdot D$, same as the base model computational cost.

