# OpenReview forum: "Position: Enough of Scaling LLMs! Lets Focus on Downscaling"
_ICML.cc/2025/Position_Paper_Track — ICML 2025 Position Paper Track poster_

### Official Review · Reviewer_rLWZ · 2025-03-10

**Significance:** 2
**Argument Clarity:** 2
**Rating:** 3
**Confidence:** 3

**Questions:**

I have included questions in the Weakness section. The main question I'd like to understand is:
Q1: whether the proposal might work at all (if so why) with training models from scratch and
Q2: why downscale the way the paper proposes instead of building small models of interest from scratch?

**Discussion Potential:**

2

**Paper Summary:**

The paper takes the position that the machine learning community may want to focus on building small models via empirical downscaling law studies. The paper proposes strategies for downscaling foundation models and datasets used to build foundation models.

**Position:**

Yes

**Position In Title:**

Yes

**Related Work:**

3

**Strengths And Weaknesses:**

# Strengths

- The paper highlights the importance of building scaling laws for small language models (SLMs).
- The paper observes that inference efficiency is an important problem. This is done via noting the importance of overtraining as well as the importance of dataset distillation-like approaches
- The paper provides a scaling law that builds on previous model pruning literature and ensemble models.

# Weaknesses

- While the problem discussed in important, the proposal appears to be vague. The proposal advocates for model ensembles and dataset distillation that are then used to train models. However, its not clear from the proposal as to whether the models will be trained from scratch or whether the "split-train-merge" pipeline in Figure 5 is used to continue training a base model.
- If the goal is to continue training from a  pretrained model via continual learning of model splits then the argument for regular scaling laws is equally important if not more so than for SLMs
- If the goal is to train from scratch, the proposal bold put forth in the paper is bold so it requires evidence to ensure validation of proposed strategies
- Since the goal of the paper is to have good SLMs, why wouldn't a typical scaling law study not apply to the regime considered in the paper. There is going to be a threshold on model size (and dataset size) below which model performance is not acceptable. Once this threshold is identified wouldn't a naive baseline be a scaling law study in the model size regime of interest?

## Minor

- There is a statement on Page 4 lines 206 - 214 that claims that small models performance can rival that of large performance. Could there be a reference included for this claim?

**Support:**

2

---

> ### Author Rebuttal · Authors · 2025-03-31
>
> We thank reviewer rLWZ for their valuable comments. Please find our responses to the queries you raised.
>
> **Clarification regarding the pipeline highlighted in Figure 5**
>
> * Our proposal involves explicitly using pre-trained models as a starting point, which are then pruned into multiple smaller models, followed by domain-adaptive continual pre-training on each pruned model, and finally ensembling them. This "split-train-merge" pipeline in Figure 5 is intended to adapt and specialize existing pre-trained models efficiently. However, the same framework can be utilized to pre-train a system of SLMs from scratch. Unlike existing LLMs, which require large-scale pre-training and domain adaptation through fine-tuning, our proposed pipeline can efficiently downscale the effective number of model parameters for domain adaptation, along with downscaling the data, therefore reducing the overall computational overhead than traditional LLM upscaling.
>
> * We argue that continual pre-training of an LLM is far less scalable than continual pre-training of a system of SLMs (our proposed pipeline). As highlighted by Kaplan and Chinchilla scaling laws, continual pre-training of an LLM only improves logarithmically with model and data size; however, the compute cost increases linearly. Therefore, it makes sense to obtain an initial checkpoint of LLM, split it into multiple smaller parts, train the smaller parts separately, and then merge them back.
>
> **Why typical scaling law is less effective than our pipeline**
>
> To demonstrate the practical viability of our approach with some examples. Suppose a pre-trained 8B model achieves a loss of 1.5. According to Kaplan's law, under a similar pre-training data size, a 1B pre-trained model can achieve $(\frac{1}{8})^{-0.08} \times 1.5 = 1.77$ loss. However, our split-train-merge pipeline achieves $1.5 + (\frac{8}{7})^{-1.1} - 0.83 + \frac{0.83}{8^{0.83}} = 1.68$ loss. Therefore, our proposed downscaling achieves better loss than the existing scaling laws for SLMs. It is important to note that all three models (pre-trained 8B, 8 pre-trained 1B, and split-train-merge model) consume the same computation.
>
> **Evidences of smaller models performing better than their larger counterparts**
>
> As demonstrated by Kondratyuk et al., 2020 and Wang et al., 2022, ensembles of smaller models can achieve higher accuracy than single larger models while using equivalent or fewer computational resources. Similarly, Lu et al., 2024 provided a comprehensive survey showing that SLMs are rapidly closing the gap with LLMs on various domains and tasks. These studies provide theoretical and empirical support for our proposed downscaling approach.
>
> **References**
>
> [1] Kondratyuk, Dan, Mingxing Tan, Matthew Brown, and Boqing Gong. "When ensembling smaller models are more efficient than single large models." arXiv preprint arXiv:2005.00570 (2020).
>
> [2] Wang, Xiaofang, Dan Kondratyuk, Eric Christiansen, Kris M. Kitani, Yair Movshovitz-Attias, and Elad Eban. "Wisdom of Committees: An Overlooked Approach To Faster and More Accurate Models." In International Conference on Learning Representations.
>
> [3] Lu, Zhenyan, Xiang Li, Dongqi Cai, Rongjie Yi, Fangming Liu, Xiwen Zhang, Nicholas D. Lane, and Mengwei Xu. "Small language models: Survey, measurements, and insights." arXiv preprint arXiv:2409.15790 (2024).

---

> > ### Comment · Reviewer_rLWZ · 2025-04-04
> >
> > Many thanks to the authors for clarifying certain points made in the paper. If I understand correctly, Figure 5 advocates for splitting a dense model to multiple smaller models and apply continual training. A reader may wonder why  i this new approach is preferable over existing methods that fall under the umbrella of "upcycling" [1, 2 and perhaps many more] dense networks to MoEs?
> >
> > [1] https://arxiv.org/abs/2212.05055
> > [2] https://arxiv.org/abs/2410.07524
> >
> > I understand authors may not be able to respond to my comment nor do I expect them to do so as its an unreasonable ask. My concern remains that I am still confused by reader to why  "ensemble-ing" should be adopted as the way to downscale. With that said, I will raise my score to 3 as the authors have done a good job clarifying my concerns. I would request the authors to update their paper to ensure this point is clear.

---

> > > ### Author Response · Authors · 2025-04-04
> > >
> > > We would like to thank the reviewer for reconsideration of the rating of our paper.
> > >
> > > Our pipeline has some of the following advantages over MOE:
> > >
> > > * **Routing Overhead Avoidance** : Krajewski et al (https://arxiv.org/abs/2402.07871) showed that “training for MOE could be bottle-necked by the routing cost". Our approach completely avoids this routing overhead.
> > > * **Incremental Upgradability**: Our ensemble can be incrementally upgraded by replacing individual models, whereas MoE architectures typically require complete retraining when modified.
> > > * **Inference Efficiency**: At inference time, naive implementations of MoE models can be up to 30 times slower than dense models due to increased memory traffic and token routing computations (https://arxiv.org/abs/2211.10017). This latency is a significant challenge for real-time applications. Our ensemble computation would be similar to the original dense model.
> > >
> > > We would also make sure the points raised by you are clearly stated in the paper. We thank you for contributing in making our paper more robust and clear

---

### Official Review · Reviewer_s6go · 2025-03-13

**Significance:** 4
**Argument Clarity:** 3
**Rating:** 4
**Confidence:** 4

**Questions:**

minor: in Figure 4, it would be more insightful if we can also add the trend of open-source LLMs there, to compare with SLMs.

**Discussion Potential:**

3

**Paper Summary:**

This paper argues limitations of scaling laws and advocates downscaling LLMs, for the sake of computational efficiency, environmental impact, and deployment constraints. Moreover, it also proposes a practical framework to downscale LLM with drastically reducing resource demands while maintaining performance.

## update after rebuttal

My concerns have been fully addressed, so I am happy to maintain my "accept" recommendation.

**Position:**

Yes

**Position In Title:**

Yes

**Related Work:**

3

**Strengths And Weaknesses:**

Strengths
- The problem addressed by this paper is important: in the field of LLMs, scaling laws are over heavily focused, which has known problems such as expensive and energy-intensive computation, substantial impact to the environment, difficult to deploy in real-world applications etc.
- This paper provides a very comprehensive overview of neural scaling laws and their criticisms.
- This paper also proposes a simple-yet-effective framework for downscaling LLMs, which looks doable and promising in practice.

Weaknesses
- It’s unclear how much performance could be reduced when downscaling LLMs (and to what scale). Some statistics could be generated by gathering the information from the ablation sections of LLM papers.

**Support:**

3

---

> ### Author Rebuttal · Authors · 2025-03-31
>
> We thank reviewer s6go for their valuable comments. Please find our responses to the query you raised.
>
> From Equation 7 in our paper, we can quantify the loss of a compressed model as - $L_{\text{compressed}} = L_0 + \left(\frac{1}{\rho}\right)^{\gamma}$, where $L_0$​ is the original model loss, $\rho$ is the sparsity ratio, and $\gamma \approx -1.1$ based on empirical findings.  When creating n model splits, the sparsity ratio becomes $\rho = 1 - \frac{1}{n}$​, which leads to
>
> $L_{\text{compressed}} = L_0 + \left(\frac{n}{n-1}\right)^{-1.1}$
>
> This formula allows us to calculate the expected performance degradation for individual pruned models. For example:
>
> * With n=8 splits, each pruned model would experience approximately a 12% increase in loss
> * With n=16 splits, each pruned model would experience approximately a 6% increase in loss
>
> However, as shown in Equation 8, when these pruned models are combined in an ensemble - $L_{\text{ensemble}} = L_0 - b + \frac{b}{n^a}$, where $b \approx 0.8$ and $a \approx 0.8$ based on Lobacheva et al., 2021. When we substitute the compressed model loss into the ensemble equation
>
> $L_{\text{ensemble}} = \left(L_0 + \left(\frac{n}{n-1}\right)^{-1.1}\right) - b + \frac{b}{n^a}$ .
>
> For $n > 8$, this ensemble loss becomes lower than the original model loss $L_0$ , providing theoretical validation for our downscaling approach. These statistics support our position that strategic downscaling can maintain performance while significantly reducing computational demands.
>
> **References**
>
> [1] Lobacheva, Ekaterina, Nadezhda Chirkova, Maxim Kodryan, and Dmitry P. Vetrov. "On power laws in deep ensembles." Advances In Neural Information Processing Systems 33 (2020): 2375-2385.

---

### Official Review · Reviewer_vYHY · 2025-03-19

**Significance:** 3
**Argument Clarity:** 2
**Rating:** 3
**Confidence:** 4

**Questions:**

Can a more explicit bridging of the scaling laws for cost (model size, emissions) and accuracy (loss) bolster the paper’s central claim?

**Discussion Potential:**

3

**Paper Summary:**

This paper advocates pivoting from ever-larger language models toward “downscaling” techniques that aim to maintain performance with less computation, environmental impact, and resource overhead. It highlights a range of emerging methods—including pruning, domain-adaptive fine-tuning, and ensembling—to show how smaller, specialized models can rival the capabilities of massive LLMs.

**Position:**

Yes

**Position In Title:**

Yes

**Related Work:**

3

**Strengths And Weaknesses:**

Strength:
- The paper systematically addresses how to reduce computation and energy usage while preserving accuracy. This focus on efficiency is timely, as many researchers have traditionally prioritized accuracy over cost and sustainability.

Weaknesses:
- The authors overlook critical efficiency techniques such as quantization and bit-precision reductions, which are widely recognized for lowering memory and energy requirements.

- Although the paper lists several scaling laws (e.g., model size, carbon emission, and loss), it lacks a cohesive link between cost models and model accuracy. A clearer framework connecting model scaling and energy/carbon costs to final accuracy would strengthen the argument.

**Support:**

2

---

> ### Author Rebuttal · Authors · 2025-03-31
>
> We thank reviewer vYHY for their valuable comments. Please find our responses to the queries you raised.
>
> * While we briefly mentioned quantization for developing efficient language models in Section 3, we intentionally focused on pruning-based approaches rather than quantization for our downscaling framework due to the computational downscaling properties of model pruning. While model pruning directly reduces the model size and, therefore, reduces the computational complexities linearly, model quantization methods do not necessarily guarantee a linear drop in computational requirements (Kumar et al., 2024). Pruning linearly affects the computational cost and CO2 emissions, but quantization, while helpful in using lesser memory, doesn’t help reduce computational footprint.
>
> * We appreciate the reviewer's suggestion to more explicitly bridge the scaling laws for the cost (model size, emissions) and accuracy (loss). This connection provides a compelling quantitative foundation for our downscaling approach. By combining the Kaplan scaling law (Equation 1 in our paper) with our CO₂ emission formula (Equation 6), we can establish a direct relationship between model performance and environmental impact:
>
> From Kaplan et al., 2020, we know that: $L(N) \propto N^{-\alpha}$, Where $L$ is the loss and $N$ is the number of parameters, with $\alpha \approx 0.08$. From Equation 6 in our paper, we established that:
>
> $\text{CO}_2\text{eq}(N, D) = (K_1 + K_2) \cdot N \cdot D \propto L^{-1/\alpha}$.
>
> Recent work by Chen et al., 2024 has shown that downstream performance $P$ can be modeled as:
>
> $P = w_1 + w_2 \cdot L$, where $w_1$ ​ and $w_2$ ​ are task-specific constants and $L$ is the pre-training loss.
>
> This leads to
>
> $\text{CO}_2\text{eq} \propto (P)^{-1/\alpha} \implies P \propto \text{CO}_2\text{eq}^{\alpha} \approx \text{CO}_2\text{eq}^{0.08}$.
>
> This demonstrates that performance improvements scale approximately with the 0.08 power of carbon emissions. Put differently, to achieve linear improvements in performance, carbon emissions must increase exponentially - a $10$\% improvement in performance would require approximately  $1.1^{12.5} = 329$% more carbon emissions. Moreover, as the model's performance increases, improving it further requires exponentially more computing.
>
> This quantitative relationship strongly reinforces our paper's central claim: traditional scaling leads to diminishing returns at exponentially increasing environmental costs. Our downscaling approach, as demonstrated in Section 4, offers a more sustainable path forward by optimizing the efficiency frontier of this relationship through targeted pruning, domain adaptation, and ensemble methods.
>
> **References**
>
> [1] Kumar, Tanishq, Zachary Ankner, Benjamin F. Spector, Blake Bordelon, Niklas Muennighoff, Mansheej Paul, Cengiz Pehlevan, Christopher Ré, and Aditi Raghunathan. "Scaling laws for precision." arXiv preprint arXiv:2411.04330 (2024).
>
> [2] Chen, Yangyi, Binxuan Huang, Yifan Gao, Zhengyang Wang, Jingfeng Yang, and Heng Ji. "Scaling laws for predicting downstream performance in LLMs." arXiv preprint arXiv:2410.08527 (2024).
>
> [3] Kaplan, Jared, Sam McCandlish, Tom Henighan, Tom B. Brown, Benjamin Chess, Rewon Child, Scott Gray, Alec Radford, Jeffrey Wu, and Dario Amodei. "Scaling laws for neural language models." arXiv preprint arXiv:2001.08361 (2020).

---

### Decision · Program_Chairs · 2025-04-30

**Decision:**

Accept (poster)

**Comment:**

This is a well-written paper championing small models instead of ever-larger models. It is well motivated; the review of scaling laws is comprehensive; the proposed solution is reasonable. The reviewers in general like the scope the paper, with some concerns on clarity. The rebuttal is strong. All the reviewers acknowledged the rebuttal or posted rebuttal comment.